# Coumarin Promotes Hypocotyl Elongation by Increasing the Synthesis of Brassinosteroids in Plants

**DOI:** 10.3390/ijms26031092

**Published:** 2025-01-27

**Authors:** Siqi Liu, Aolin Ma, Jie Li, Zhixuan Du, Longfei Zhu, Guanping Feng

**Affiliations:** Key Laboratory of Jiangxi Province for Biological Invasion and Biosecurity, School of Life Sciences, Jinggangshan University, Ji’an 343009, China; sqliujgsu@163.com (S.L.); alimajgsu@163.com (A.M.); lijie040828@163.com (J.L.); du_zhixuan@163.com (Z.D.); longfzhu@163.com (L.Z.)

**Keywords:** *Arabidopsis thaliana*, coumarin, hypocotyl, brassinosteroid, cell elongation

## Abstract

Coumarins are natural products commonly found in plants and are typical allelopathic substances that strongly affect the growth of plants after being exudated from the root and help plants absorb Fe in cases of iron deficiency. Although coumarins have been found to have multiple effects, this understanding is still relatively limited. Here, we show that coumarin significantly promotes the elongation of the hypocotyl by enhancing cell elongation. Further research has found that coumarin increases the content of BR in plants by enhancing the expression of brassinosteroid (BR) synthesis genes. The effect of coumarin on promoting hypocotyl elongation is completely blocked by the mutation of the BR synthesis gene *DEETIOLATED 2* (*DET2*) or the co-addition of the BR synthesis inhibitor brassinazole (BRZ). Genetic analysis using Arabidopsis mutants showed that coumarin promoting hypocotyl elongation depends on the signaling pathway of the BRs. Overall, coumarin promotes elongation of the hypocotyl by increasing the synthesis of BRs in plants. These results provide us with new insights into the role of coumarins and offer strong theoretical support for the mechanisms of interactions between plants.

## 1. Introduction

Allelopathy is a phenomenon where plants release certain organic compounds into their environment that inhibit the germination and growth of their offspring and other nearby plants. Coumarins have been identified as the most prevalent allelochemicals among certain plant species, exerting the most pronounced effects on autotoxicity [1,2,3]. Coumarins are a group of natural products that are widely found across various plants, and have been detected in over 330 genera from 74 families [4]. Numerous coumarin-based compounds have been extracted and identified from a range of plant parts including the stems, leaves, flowers, seeds, roots, and exudates of the roots [4,5]. As specialized metabolites originating from phenylpropanoid biosynthesis, coumarins are distinguished into five categories, each defined by its unique substituent architecture [6]. Simple coumarins have emerged as the most prevalent class of coumarins that have been isolated and characterized in various plant species.

Coumarins are synthesized via feruloyl-CoA 6-hydroxylase1 (F6′H1) and are excreted into the rhizosphere by the iron deficiency-regulated ABC transporter pleiotropic drug resistance 9 (PDR9) [7,8]. In recent research, the pivotal function of coumarin exudation in enhancing iron acquisition by plants has been established [8,9,10,11]. Transcription factor MYB63 dictates the synthesis of coumarin by modulating the activity of COSY and F6′H1, and facilitates its secretion via PDR9 under the duress of simultaneous Fe and P scarcity [12]. Coumarin exudation, regulated by MYB72, plays a pivotal role in sculpting the assembly of the root microbiome, which in turn promotes overall plant well-being [13]. Coumarin exerts an inhibitory effect on germination and reduces the growth of lettuce seedlings [14]. Yao’s investigation into Italian ryegrass revealed that coumarin disrupted the cell membrane structure in the seed endosperm cells, and that this disruption notably hampered seed germination and the early development of seedlings [15]. Coumarin markedly reduced the germination rate of alfalfa seeds as well as impacted their germination potential, radicle length, and overall germ length [16,17].

Allelochemicals affect the growth and development process of plants by altering the content or activity of hormones. Coumarins can also affect the plant growth and development of plants by changing or disrupting the hormonal balance. The study conducted by the researcher explored the effects of coumarin on annual ryegrass, revealing that coumarin suppressed the production of gibberellin and auxin while enhancing the secretion of abscisic acid, thereby modifying the plant’s internal hormone balance [18]. The application of coumarin to alfalfa seedlings induced the upregulation of key genes such as *NCED*, *ZEP*, and *BG*, which are integral to the abscisic acid biosynthetic pathway, leading to an elevation in the levels of abscisic acid within the alfalfa root system [19]. The coumarin-induced delay of rice seed germination is mediated by the suppression of abscisic acid catabolism and the production of reactive oxygen species [20].

In order to investigate the molecular mechanism of coumarin allelopathy, we attempted to screen out mutants with differential responses to coumarin, hoping to discover specific genes involved in coumarin action. Unexpectedly, we found that coumarin had a significant effect on promoting hypocotyl growth. Further research suggests that coumarin promotes hypocotyl growth by promoting the biosynthesis of brassinosteroids in plants.

## 2. Results

### 2.1. Coumarin Promotes Elongation of Hypocotyl in Plants

Coumarin, as a well-known allelopathic compound, strongly affects the seed germination and seedling growth of plants. In addition, in iron-deficient environments, coumarin also has the function of helping plants absorb iron. Next, our experiments found that coumarin had a significant promoting effect on the elongation of plant hypocotyls at low concentrations. When 50 µM of coumarin was added to 1/2MS medium, the hypocotyl length of plants such as alfalfa (*Medicago sativa*), bok choy (*Brassica rapa*), sesbania (*Sesbania cannabina*), and tomato (*Solanum lycopersicum*) increased by 155%, 52%, 64%, and 38%, respectively, compared with the control plants (Figure 1). Coumarin had a significant promoting effect on the elongation of hypocotyls in various plants, although the magnitude of increase varied among different plant species. The results clearly indicate that coumarin has the function of promoting plant hypocotyl elongation.

As expected, coumarin significantly promoted the growth of hypocotyls in the model plant *Arabidopsis thaliana*. In the 1/2MS medium containing 50 µM coumarin, the hypocotyls of Arabidopsis were nearly twice as long as those of the control plants without the addition, once again confirming the promoting effect of coumarin on hypocotyl elongation (Figure 2A,C). In order to clarify the cytological reasons for coumarin promoting hypocotyl elongation, we conducted cytological observations of Arabidopsis hypocotyls and found that coumarin promoted hypocotyl elongation due to its promoting effect on cell elongation in the hypocotyl (Figure 2B,D). The dose–response curve showed that coumarin could promote the elongation of hypocotyls at a concentration of 10 µM. The promoting effect increased with the increase in coumarin concentration, and reached its peak at a concentration of 80 µM (Figure 2E). These results indicate that coumarin promotes hypocotyl elongation by promoting cell elongation.

### 2.2. Mutant DET2 Is Insensitive to Coumarin

In order to investigate the mechanism of coumarin promoting hypocotyl elongation, we extensively screened mutants related to the synthesis and signal transduction of various plant hormones. We found that the loss-of-function mutant of *DEETIOLATED 2* (*DET2*), a key gene for BRs synthesis, showed no response to coumarin, indicating that coumarin promoting hypocotyl elongation may depend on the biosynthesis of BRs (Figure 3A,B). Next, we used quantitative real-time PCR (qRT-PCR) to detect the expression levels of DET2 and other important genes involved in BR synthesis after coumarin treatment and found that the expression of *DET2* and *constitutive photomorphogenic dwarf* (*CPD*) was significantly induced by coumarin (Figure 3C). These results strongly indicate that coumarin promotes hypocotyl elongation by inducing the synthesis of BRs in plants.

### 2.3. Coumarin Promotes Hypocotyl Elongation Depends on the Synthesis of BRs

In order to further clarify the correlation between coumarin promoting hypocotyl elongation and BR biosynthesis, we used HPLC-MS/MS to detect the concentration of brassinolide (BL) before and after coumarin treatment. Coumarin treatment increased the BL in the plants from 0.019 ng/g to 0.054 ng/g, with an increase up to 184% (Figure 4). Previous studies have clearly shown that BRs have a significant promoting effect on cell elongation and play an important role in the elongation of plant hypocotyls. Coumarin treatment led to a significant increase in the concentration of BRs in plants, which is likely to be the direct cause of its promotion of hypocotyl elongation.

To further confirm that coumarin promoting hypocotyl elongation was dependent on BR synthesis, we added epi-brassinolide (eBL), the most active natural BR, and the brassinolide synthesis inhibitor brassinazole (BRZ) with coumarin, and found that the addition of 5 μM eBL and 50 μM coumarin could slightly increase the phenotype of coumarin promoting hypocotyl elongation (Figure 5). This may be due to the fact that the addition of both increased the abundance of brassinolide in the plant. When 1 μM BRZ was added together with coumarin, the effect of coumarin on promoting hypocotyl elongation was completely blocked, indicating that the presence of BRZ led to the complete inhibition of coumarin’s promotion of BR synthesis in the plants, and the effect of coumarin on promoting hypocotyl elongation was completely inhibited (Figure 5). These experimental results clearly indicate that the effect of coumarin on promoting hypocotyl elongation is entirely dependent on the biosynthesis of BRs in plants.

### 2.4. Coumarin Promotes Hypocotyl Elongation Through BR Signaling

To further investigate the relationship between the coumarin promoting hypocotyl elongation and BRs, we analyzed the response of the BR receptor protein BRI1 and major signaling factors to coumarin. The results showed that the hypocotyl length of the mutant *bri1-5*, loss-of-function of BRI1, was shorter than that of the wild-type, while the hypocotyl length of the *brz1-1D*, gain-of-function mutant of BR key signaling protein, brassinazole-resistant 1 (BZR1), was significantly longer than that of Col, indicating that inhibition or enhancement of the BR signal led to changes in the hypocotyl length (Figure 6A). Both coumarin and eBL could promote the elongation of the hypocotyls in mutants *bri1-5* and *brz1-1D*, while the combined addition of both led to a much smaller increase in hypocotyls than the sum of their respective promotion of the hypocotyls, indicating an overlap between the effects of coumarin and BRs (Figure 6A). However, the brassinolide synthesis inhibitor brassinazole (BRZ) completely blocked the function of coumarin and eBL in promoting hypocotyl elongation (Figure 6A). The gene expression levels of brassinosteroid receptor protein BRI1 and other key signaling factors were detected using qRT-PCR, and it was found that the expression of *BRI1* and *BRZ1* was induced by coumarin (Figure 6B). After coumarin treatment, the expression level of *BRI1* and *BRZ1* increased to 143% and 195% of the control, respectively (Figure 6B). The expression levels of other BR signaling genes, such as *BRI1-associated receptor kinase 1*(*BAK1*), *brassinosteroid-insensitive 2* (*BIN2*), *BRI1-EMS-suppressor 1* (*BES1*), showed no significant difference compared with the control after coumarin treatment. These experimental results demonstrate that the effect of coumarin on promoting hypocotyl elongation is consistent with the role of BRs, and that coumarin relies on BR signaling to exert its effect on promoting hypocotyl elongation.

## 3. Discussion

### 3.1. Promoting Hypocotyl Elongation Is Another New Effect of Coumarin

Coumarins play multiple roles in regulating plant growth and development, functioning as plant signaling agents and phytoalexins, but they also possess a range of therapeutic applications including anticancer, antiviral, blood sugar-lowering, blood pressure-reducing, and neuroprotective effects [4]. Nevertheless, the synthesis and secretion of coumarins not only have an allelopathic impact on neighboring vegetation, but also suppress the germination of seeds and the development of seedlings of the plants’ own progeny, demonstrating a significant degree of autotoxicity [21]. Coumarins notably suppress the activity of superoxide dismutase (SOD), peroxidase (POD), and catalase (CAT) in the alfalfa seedlings, which consequently lead to a marked elevation in malondialdehyde (MDA) [22]. Plants increase iron absorption by secreting coumarins under iron deficiency conditions, but also produce and accumulate different types of coumarins when encountering pests and diseases [23,24]. Treatment with low concentrations of coumarin showed minimal impact on the chlorophyll levels in canola (Brassica campestris) plants; however, at high concentrations, there was a marked decrease in seedling biomass as well as a substantial reduction in the chlorophyll content [25]. The application of varying concentrations of coumarin solution to annual ryegrass seedlings led to an elevation in starch content and the development of starch granule complexes [19,26]. As a natural product synthesized and secreted by plants, coumarins have multiple functions and play important roles in enhancing plant adaptation to the surrounding environment. Our findings have identified another important effect of coumarin, which promotes hypocotyl elongation and affects their adaptability to complex and changing environments.

### 3.2. Coumarin Promotes Hypocotyl Elongation by Inducing BR Synthesis

The hypocotyl serves as the embryonic stem linking the primary root to the cotyledons, and the length of the hypocotyl varies extensively under different conditions [27]. The elongation of the hypocotyl in both light-exposed and dark-grown seedlings depends on cellular expansion, particularly of the epidermal cells. The most prominent hormones required for hypocotyl elongation in darkness are auxin, gibberellic acid (GA), and brassinosteroids (BRs) [28,29,30]. Brassinosteroids, a class of steroidal phytohormones that govern various developmental pathways in plants, are recognized for their role in enhancing plant cell elongation by facilitating the relaxation of cell walls [31]. Preserving the homeostasis of brassinosteroids is thus vital for the optimal functioning of these hormones in plants. The biosynthesis of brassinosteroids encompasses parallel and intricately interconnected pathways that result in the formation of C27, C28, and C29 steroids, classified according to their carbon atom count [32]. The transcriptional effects of BRs are channeled through the transcription factors BZR1 and BRI1-EMS-suppressor 1 (BES1), which become derepressed following the inactivation of their suppressor, brassinosteroid-insensitive 2 (BIN2). Among the genes targeted by BZR1 and BES1, some are directly involved in the elongation process and encompass factors that regulate the cytoskeleton and cell wall dynamics [31,33]. Coumarin, as a typical allelopathic species, can affect the growth of other surrounding plants, but its receptors have not been identified yet. Our research found that coumarin can significantly promote the elongation of the hypocotyl, and the experimental results showed that coumarin’s promotion of hypocotyl elongation depends on its induction effect on BR synthesis. Further mutant analysis and gene expression analysis showed that coumarin acts on the key gene *DET2*, involved in BR synthesis. Although BRs are important regulators of growth, there is limited evidence to suggest environmental control over their concentrations [27]. By remodeling cell walls, BRs reduce the epidermal constraint and control growth coordination between layers by means of mechanics [34]. Perhaps the effect of coumarin on promoting hypocotyl elongation is due to this mechanism of action of BRs. BRs and auxin act cooperatively to promote cell elongation in numerous species and promote hypocotyl elongation under many conditions. Arabidopsis Aux/IAA gain-of-function mutants have displayed reduced BR sensitivity in hypocotyl elongation assays, and BR treatment enhances shoot polar auxin transport in plants [35]. The effect of coumarin on promoting hypocotyl elongation may require this cooperative relationship between BR and auxin. Our research findings indicate that plants can regulate the levels of BRs by altering the secretion of coumarins, thereby controlling the length of the hypocotyls. This provides conclusive evidence that plants alter the concentration of BRs to cope with environmental changes.

### 3.3. The Effect of Coumarin on Promoting Hypocotyl Elongation Increase Plant Adaptability

Plants do not always inhabit optimal environments and frequently encounter various forms of environmental stress throughout their growth and development. In order to adjust to changing environmental conditions, plants synthesize specialized metabolites as a response to these stimuli, which in turn influence the expression of genes involved in biosynthetic pathways [36,37]. The biosynthesis of coumarins can be triggered by a range of environmental conditions. This process includes intricate glycosylation modifications that can aid in adapting to environmental shifts by enhancing the stability and biological efficacy of coumarins [38,39]. Under iron-deficient conditions, plant roots exude coumarins including scopoletin, esculetin, and daphnetin [10]. Salt-alkaline stress significantly enhances the accumulation of coumarins such as scopoletin and scopolin in L. barbarum [40]. Additionally, the accumulation of scopoletin has been noted in cells subjected to low-temperature conditions [41]. Furthermore, coumarins including scopoletin, scopolin, umbelliferone, and strigolactone fulfill significant ecological functions in the adaptation of plants to drought conditions [42]. It was observed that drought stress significantly elevated the coumarin concentration in the leaf extracts of *Ficus deltoidea.* The legume *Melilotus albus* contains abundant coumarins and demonstrates a high level of resilience to harsh environmental conditions including drought, cold, and salinity [43]. The elongation of the hypocotyl is strongly influenced by the surrounding environmental conditions. Under different environmental conditions, plant seedlings can gain competitive advantages through faster hypocotyl elongation. Coumarin, which rapidly expresses in response to different environments, may play an undiscovered role in these processes.

## 4. Materials and Methods

### 4.1. Plant Materials and Growth Conditions

For this investigation, the seeds of alfalfa (*Medicago sativa*), bok choy (*Brassica campestris* L. ssp. *chinensis Makino*), sesbania (*Sesbania cannabina*), tomato (*Solanum lycopersicum*), and *Arabidopsis thaliana* Columbia-0 (Col) were used. The *Arabidopsis* mutants of *det2*, *brz1-1D*, and so on were purchased from NASC. After sterilization with 2% NaClO for 10 min and rinsed three times with sterile water, the seeds were planted onto 1/2 MS medium supplemented with 1% sucrose and 0.7% agar. Subsequently, the plate was placed in a growth chamber maintained at a temperature of 22 ± 1 °C, illuminated at a flux of 80–90 μmol m^−2^ s^−1^, and subjected to a photoperiod of 16 h of light followed by 8 h of darkness, which facilitated seed germination and the subsequent growth of seedlings. Coumarin (CAS: 91-64-5) and BRZ (CAS: 91-64-5) were purchased from Aladdin (Aladdin-Holdings Group, Beijing, China), and eBL (CAS: PH161) from Coolaber (Coolaber Science & Technology, Beijing, China).

### 4.2. Cytological Observation of Arabidopsis Hypocotyl

Arabidopsis seedlings grown vertically in 1/2MS medium for 7 days were used for the hypocotyl cytology observations. Arabidopsis seedlings were immersed in a 75% ethanol solution and gradually increased to 100% ethanol solution for decolorization treatment. The processed hypocotyl was placed on a glass slide, transparent liquid HCG (24 g chloral hydrate, 3 mL glycerol, and 9 mL H_2_O) was added, and then covered with a cover glass. Finally, the cells of the hypocotyl were observed under a microscope and photographs taken for analysis. Using ImageJ software, we measured the length of the hypocotyl and the epidermal cells.

### 4.3. Gene Expression Analysis

RNA was extracted from the hypocotyl of the 7-day-old seedlings using the TRIzol reagent (Invitrogen, USA). The extracted RNA was then converted into cDNA using SuperScript-III Reverse Transcriptase (Invitrogen) for further analysis. Quantitative real-time PCR (qRT-PCR) was carried out on an QuantStudio 3 thermal cycler, utilizing the SYBR^®^ Premix Ex TaqTM Kit (Takara, Dalian, China) as per the manufacturer’s protocol. The *actin2* gene was used as the internal control. The expression levels of each gene were normalized and expressed relative to the control, based on three biological replicates. The data are presented as the mean ± standard error (SE); statistical significance was determined by the Student’s *t*-test: ** indicates *p* < 0.05; *** indicates *p* < 0.01. The primers employed for the expression analysis are listed in Appendix A.

### 4.4. Determination of the Concentration of Brassinolide

Arabidopsis seedlings grown for 10 days in 1/2MS medium supplemented with 50 micromolar coumarin and a blank control were used to determine the concentration of brassinolide. Approximately 2 g of Arabidopsis seedlings were quickly frozen in liquid nitrogen and ground into powder, then placed in a centrifuge tube. We added 10 mL of 95% methanol precooled to 4 °C, then centrifuged and collected the supernatant. Next, 4 µL of an internal standard solution with a concentration of 1 μg/mL was added to the mixture, purified using an MCX solid-phase extraction column, and eluted with 5 mL of methanol. The eluent was evaporated and dried under a nitrogen flow, then dissolved again in 200 µL of methanol, filtered through a 0.22 µm membrane, and analyzed by an HPLC-MS/MS system.

## 5. Conclusions

In this study, we found that coumarin, as a typical allelopathic substance, not only affects plant seed germination and growth and development, but also has the effect of promoting hypocotyl growth. We investigated the effect of coumarin in promoting hypocotyl growth using the model plant Arabidopsis. It was found that coumarin promoted cell elongation in the hypocotyl by inducing the biosynthesis of brassinosteroids in the plant. When the key gene *DET2* for brassinosteroid synthesis in Arabidopsis was functionally deficient or treated with the brassinosteroid synthesis inhibitor BRZ, the effect of coumarin on promoting hypocotyl elongation was completely blocked. These research results have provided us with new insights into the role of coumarins in plant growth and development as well as their interactions with the surrounding environment.

## Figures and Tables

**Figure 1 ijms-26-01092-f001:**
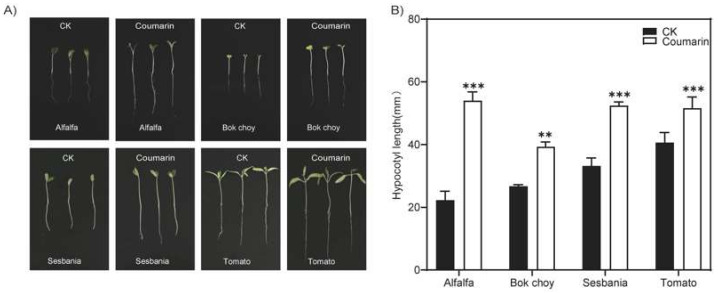
Coumarin promotes hypocotyl elongation in plants. Phenotypes (**A**) of the hypocotyl of the 10-day-old seedlings growing and on 1/2MS medium containing 50 µM coumarin. CK, the blank control. The length of the hypocotyl (**B**) of the seedling in (**A**). Student *t*-test: **, *p* < 0.05; ***, *p* < 0.01.

**Figure 2 ijms-26-01092-f002:**
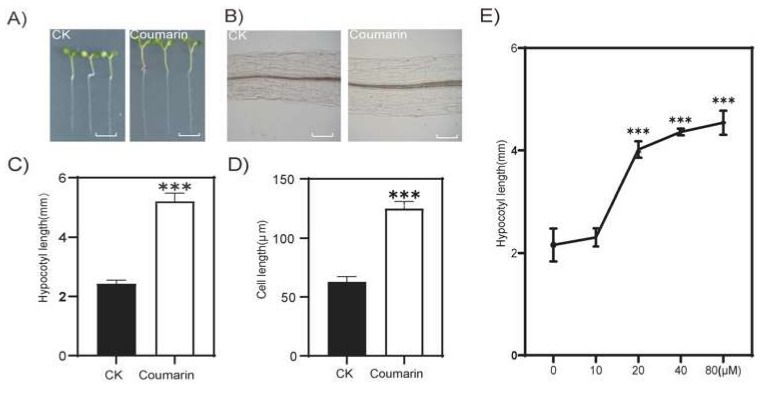
Coumarin promotes hypocotyl elongation by promoting cell expansion. Phenotypes (**A**) and C = cytological observation of the hypocotyl (**B**) of the 7-day-old Arabidopsis seedlings growing and on 1/2MS medium containing 50 µM coumarin. CK, the blank control. Scale bar, 0.5 mm (**A**) and 100 µm (**B**). The length of the hypocotyl (**C**) of the seedling in (**A**) and the cell length of the hypocotyl (**D**) of the seedling in (**A**). The dose–response curve of coumarin (**E**). Student *t*-test: ***, *p* < 0.01.

**Figure 3 ijms-26-01092-f003:**
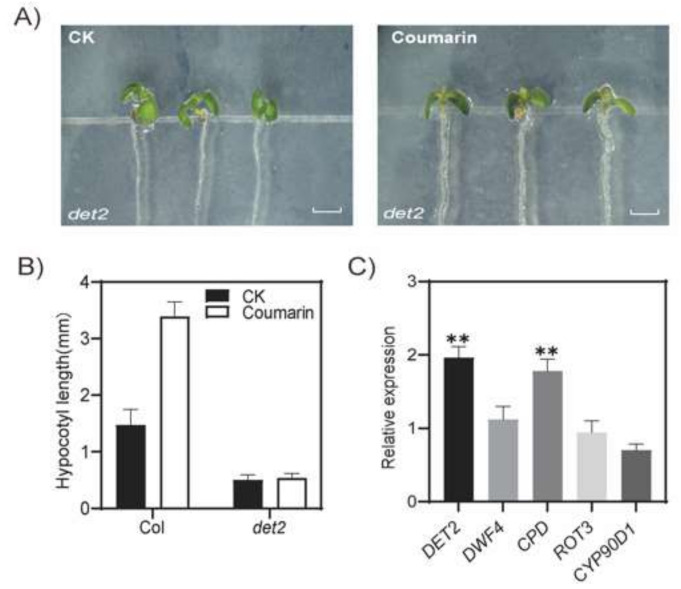
Mutant *det2* is insensitive to coumarin. Phenotypes (**A**) and the length (**B**) of the hypocotyl of the 7-day-old *det2* mutant growing on 1/2MS medium containing 50 µM coumarin. CK, the blank control. Col, Arabidopsis wild-type Columbia. Scale bar, 0.5 mm. Relative expression of genes involved in BR synthesis (**C**). *DET2*, *de-etiolated 2*; *DWF4*, *dwarf 4*; *CPD*, *constitutive photomorphogenic dwarf*; *ROT3*, *rotundifolia 3*; *CYP90D1*, *cytochrome P450*, *family 90*, *subfamily D*, *polypeptide 1*. Student *t*-test: **, *p* < 0.05.

**Figure 4 ijms-26-01092-f004:**
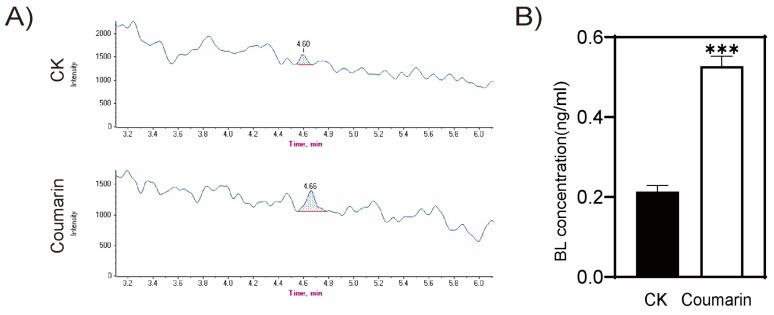
HPLC-MS/MS was used to detect the concentration of brassinolide (BL) before and after coumarin treatment. HPLC-MS spectrum (**A**) and BL concentration (**B**). CK, the blank control. Student *t*-test: ***, *p* < 0.01.

**Figure 5 ijms-26-01092-f005:**
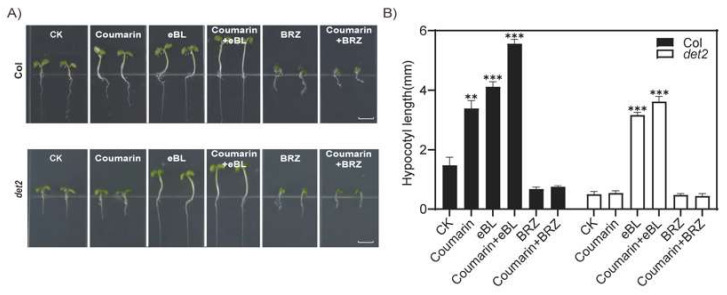
Coumarin promotes hypocotyl elongation, which is dependent on the BRs. The hypocotyl length of Col and *det2* growing in the 1/2MS medium, adding coumarin with eBL or BRZ (**A**) and the length of hypcotyl (**B**). Student *t*-test: **, *p* < 0.05; ***, *p* < 0.01.

**Figure 6 ijms-26-01092-f006:**
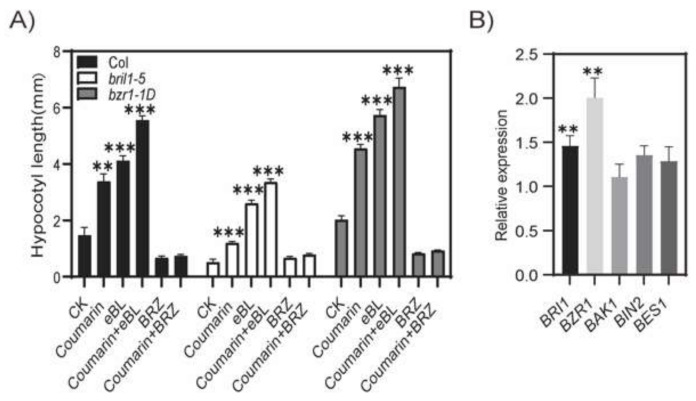
Coumarin promotes hypocotyl elongation through BR signaling. The hypocotyl length of Col, *bri1-5*, and *bzr1-1D* growing in the 1/2MS medium, adding coumarin with eBL or BRZ (**A**). Relative expression of genes involved in BR signaling after coumarin treatment (**B**). qRT-PCRs were performed with RNAs isolated from the leaves of 7-d-old plants. Data are shown as the mean ± SE for three biological replicates. *ACTIN2* used as the internal control. Student *t*-test: **, *p* < 0.05; ***, *p* < 0.01.

## Data Availability

The data presented in this study are available on request from the corresponding author.

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
