# Peer review of "Coumarin Promotes Hypocotyl Elongation by Increasing the Synthesis of Brassinosteroids in Plants"

_ijms, 2025, doi:10.3390/ijms26031092_

Round 1

Reviewer 1 Report

Comments and Suggestions for Authors

In the current paper authors studied effects of coumarin and det2 mutation on hypocotyl elongation under light conditions in the presence of exogenous carbon source.

The results are interesting, but text require corrections and re-evaluation of some points.

Auxin is the main regulator of hypocotyl elongation and certain auxin biosyntesis pathway (YUC1D) led to hypocotyl elongation under light conditions.  Brassinosteroid can serve as modulator of auxin synthesis/distribution what can be confirmed by your images with root growth.  

The most significant coumarin effcet is on root, whch directly link wit sink-sourec effcet of BR.

Need to be pointed out.

In will be great to describe localization of BR syntnesis: hypocotyl? Leaf? Cotyledon?

Line 91: „The concentration gradient“ = dose-response curve.

Line 92: “coumarin showed that coumarin“ ???

Fig 2A: coumarin significantly inhibited root growth. Maybe shoot-derived auxin accumulated in hypocityl (not move to root) and induce elongation?

Line 168: “the behavior of BRs“ ??

Line 198: „function of coumarin“ – it is not a function, it is effect.

Line 205: „particularly of epidermal cells.” ¿? In ideal case you need to label cell border and made 3D reconstruction.

Line 218: “significant hormones” ¿?? Hormines can not be significant itself.

Lines 248 – 249_ please, clarify seeds type of alflafa, tomato etc. Is it any specifc lines?

Line 260: „Each concentration level“ ??? Level = redundant.

Line 261: “total of three operations” ¿?

Line 278: “Weigh an appropriate amount of sample“ ?? Are you really weigh powder after grinding?

Line 287: “We conducted in-depth molecular mechanism research on the function of coumarin in promoting hypocotyl growth using the model plant Arabidopsis.“ ??  RNA was isolated from whole seedlings including cotyledon, young leaf, root.

Line 290 – 292_ grammar, corrections are required.

Comments on the Quality of English Language

scientific english require corrections. There is no "significant hormone" etc.

Author Response

Comments 1: Auxin is the main regulator of hypocotyl elongation and certain auxin biosyntesis pathway (YUC1D) led to hypocotyl elongation under light conditions.  Brassinosteroid can serve as modulator of auxin synthesis/distribution what can be confirmed by your images with root growth.  

Response 1: Auxins play an important regulatory role in the elongation of plant hypocotyls. We tested the content of auxin under coumarin treatment, and the results showed that the auxin content was not affected by coumarin. In addition, we have also verified that the response of the mutations of YUC1 and YUC9, and yuc2/5/8/9 to coumarin , are consistent with the control. Therefore, coumarin in promoting hypocotyl elongation is mainly directly related to BR synthesis.

Comments 2: The most significant coumarin effcet is on root, whch directly link wit sink-sourec effcet of BR. Need to be pointed out.In will be great to describe localization of BR syntnesis: hypocotyl? Leaf? Cotyledon?

Response 2: The apical meristem of plants, including the stem tip and root tip, is an important site for brassinosteroid synthesis. The active cell division and growth in these regions require the involvement of brassinolide. Brassinosteroids (BRs) are different from other plant hormones such as auxins in that their transport within the plant body is not polar. The transport of brassinosteroids is considered non-polar, which means they can move in various directions within the plant body without being restricted by specific directions. In the discussion section, we added the effect of coumarin on the synthesis site and transport of BR.

Comments 3: Line 91: „The concentration gradient“ = dose-response curve.

Response 3:  Corrected

Comments 4: Line 92: “coumarin showed that coumarin“ ???

Response 4: Corrected

Comments 5: Fig 2A: coumarin significantly inhibited root growth. Maybe shoot-derived auxin accumulated in hypocityl (not move to root) and induce elongation?

Response 5: We tested the content of auxin under coumarin treatment, and the results showed that the auxin content was not affected by coumarin. We are currently testing whether the auxin content in the hypocotyl is affected. But due to the given deadline for modification, we have not yet completed this experiment.

Comments 6: Line 168: “the behavior of BRs“ ??

Response 6: the role of BRs

Comments 7: Line 198: „function of coumarin“ – it is not a function, it is effect.

Response 7: Corrected

Comments 8: Line 205: „particularly of epidermal cells.” ¿? In ideal case you need to label cell border and made 3D reconstruction.

Response 8: We haven't mastered the technology of making 3D reconstructions yet. Very sorry!!!

Comments 9: Line 218: “significant hormones” ¿?? Hormines can not be significant itself.

Response 9:  Corrected to “the important regulator of growth”

Comments 10: Lines 248 – 249_ please, clarify seeds type of alflafa, tomato etc. Is it any specifc lines?

Response 10:  Added Latin names for seeds

Comments 11: Line 260: „Each concentration level“ ??? Level = redundant.

Response 11: Corrected

Comments 12: Line 261: “total of three operations” ¿?

Response 12: Corrected to “Three repetitions”

Comments 13: Line 278: “Weigh an appropriate amount of sample“ ?? Are you really weigh powder after grinding?

Response 13: Weigh 2 grams of seedlings

Comments 14: Line 287: “We conducted in-depth molecular mechanism research on the function of coumarin in promoting hypocotyl growth using the model plant Arabidopsis.“ ??  RNA was isolated from whole seedlings including cotyledon, young leaf, root.

Response 14: RNA was isolated from whole seedlings

Comments 15: Line 290 – 292_ grammar, corrections are required.

Response 15: Corrected

Reviewer 2 Report

Comments and Suggestions for Authors

The paper has merit, but some aspects require major improvement. My main concern has to do with figure 6B. In the legend, it is not even mentioned which is the experiment. I guess (as nothing is described in the legend) that this is the relative expression of several genes quantified by RT PCR. Please describe the expression, the n number for each bar, and the statistical test that has been used. Which value has been used as a reference? as usual is 1. Please mention this.

In the materials section, it is not stated which genes have been used as controls for normalization, and it refers to a primer list in the supplemental material, but there is not supplemental material in the submission. 

Discuccion: authors have decided to submit the manuscript to a molecular journal, but the weak point is that they do not have a molecular mechanism. Which is the coumarin receptor in the plant? With which protein interacts? Is there anything known? Authors should explain well this point in the discussion to justify the publication in IJMS rather than other journals devoted to plant biology and physiology such as "plants.". 

Figures 1A and 5a: Please enhance resolution. Nothing can be seen.

Line 233: Please correct the citation format. 

Author Response

Comments 1: The paper has merit, but some aspects require major improvement. My main concern has to do with figure 6B. In the legend, it is not even mentioned which is the experiment. I guess (as nothing is described in the legend) that this is the relative expression of several genes quantified by RT PCR. Please describe the expression, the n number for each bar, and the statistical test that has been used. Which value has been used as a reference? as usual is 1. Please mention this.

Response 1: Specific descriptions of gene expression changes after coumarin treatment have been added to the manuscript, and experimental data processing instructions have been included in the legend.

Manuscript: After coumarin treatment, the expression level of BRI1 and BRZ1 increased to 143% and 195% of the control (Figure 6B). The expression levels of other BRs signaling genes, BRI1-ASSOCIATED RECEPTOR KINASE 1(BAK1), BRASSINOSTEROID-INSENSITIVE 2 (BIN2), BRI1-EMS-SUPPRESSOR 1(BES1), showed no significant difference compared to the control after coumarin treatment.

Legend:  qRT-PCRs were performed with RNAs isolated from the leaves of 7-d-old plants. Data are shown as mean ± SE for three biological replicates.ACTIN2 used as internal control.

Comments 2: In the materials section, it is not stated which genes have been used as controls for normalization, and it refers to a primer list in the supplemental material, but there is not supplemental material in the submission. 

Response 2: ACTIN2 used as internal control. supplemental material is uploaded again.

Comments 3: Discuccion: authors have decided to submit the manuscript to a molecular journal, but the weak point is that they do not have a molecular mechanism. Which is the coumarin receptor in the plant? With which protein interacts? Is there anything known? Authors should explain well this point in the discussion to justify the publication in IJMS rather than other journals devoted to plant biology and physiology such as "plants.". 

Response 3: Coumarin, as a typical allelopathic species, can affect the growth of other surrounding plants, but its receptors have not yet been identified. Our research found that coumarin can significantly promote the elongation of the hypocotyl, and the experimental results showed that coumarin's promotion of hypocotyl elongation depends on its induction effect on BR synthesis. Further mutant analysis and gene expression analysis showed that coumarin acts on the key gene DET2 involved in BR synthesis. We have supplemented these contents in the discussion section.

Comments 4: Figures 1A and 5a: Please enhance resolution. Nothing can be seen.

Response 4: Replaced with higher resolution images.

Comments 5: Line 233: Please correct the citation format. 

Response 5: Corrected

Round 2

Reviewer 1 Report

Comments and Suggestions for Authors

Thank you for response.

The text have some more clarity, but some previous commnets still not became clear. Please, look on it more deep, My best regards!

Response 1: Auxins play an important regulatory role in the elongation of plant hypocotyls. We tested the content of auxin under coumarin treatment, and the results showed that the auxin content was not affected by coumarin. In addition, we have also verified that the response of the mutations of YUC1 and YUC9, and yuc2/5/8/9 to coumarin , are consistent with the control. Therefore, coumarin in promoting hypocotyl elongation is mainly directly related to BR synthesis.“

If you want to tell about BR role/effect, you need to use „auxin-free“ system only. Moreover, if you measure so-called „auxin contents“ (balance between synthesis, transport, degradation, conjugation“ in elongated hypocotyl, of course, no differences will be detected. Auxin accumulation occurred before elongation. Please, next time use kinetimatical study, not after process already completed.

Comments 2: The most significant coumarin effect is on the root, which directly link wit sink-source effect.

The effect of coumarin on the root did not described. One can not took one part of the whole organism and ignore effect on the others. I can expect that coumarin prevent flux to root and induce elongation of the hypocotyl (re-distribution of metabolites). This porcess can be proven only in kinematic study. Once process complete, resources rich balance. But pulses of resources come before elongation or at the early stages.  

Both points need to be mentioned in discussion.

Some minor comments:

Line 108: “Coumarin elongates of Arabidopsis hypocotyl by promoting cell elongation.” ¿?  Scientific grammar, please.

Line 114: “we used quantitative real-time PCR (qRT-PCR)” ¿?  Is it RNA from hypocotyl of whole seedlings?

Line 156: „hypcotyl“ ?#

Line 194: „new function of coumarin“ ?? Coumarin have not a function, it can be effect.

Lines  215, 246: „help plants improve their adaptability to complex and changing en- 215

vironments.“ – how long hypocotyl and short root can improve adaptivity?

Line 276: why did you add sucrose as extra playerin hypocotyl elongation? Next time you need to avoid it.

Line 277: “Each concentration is treated for about 1 hour, with a total of three repetitions.“ Very unclear. Seedlings, but not concentration can treated.

Line 296: RNA form whole seedlings is a problem. You mention only specific organ, and each organ/cell type have own mRNA profile.

Line 321: „conducted molecular mechanism research” ¿? There is not  such research. Please, use scientific formulation.

Author Response

(The authors gave the same response as above.)

Reviewer 2 Report

Comments and Suggestions for Authors

The paper has been dramatically improved, and all my concerns have been resolved, so I can be happy to publish. 

I would only like to point out that the authors use a single gene (ACTIN1) as a reference for the qRT-PCR experiments. Please note that many journals no longer accept a single gene as a reference, and two or three must be used to grant the robustness of the data. Consider this in future projects.

Author Response

(The authors gave the same response as above.)

Round 3

Reviewer 1 Report

Comments and Suggestions for Authors

Thank you. However, I have not seen response to comments. Only repetion from previous one.

Here is one require answers:

Point 1: it seems authors measure hormone „concentartion“ at one time point. However, this is wrong: auxin peak preceed elongation and to tell about role of local auxin effcet several time points is required. Moreover, conclusion about brasssinosteroids require as control auxin-free system.

Line 288: Total RNA from 7 days old seedlings can not provide ypu data about molecular mechanisms. Each cell type have own RNA profile and in the case of total RNA it is a compilation of many different profiles, but elongation is a cell-type specific gene expression (like vacuolar transporter etc). This need to be mentioned in discussion.  

Line 312: the function of promoting hypocotyl elongation“ ?? Function = effect.

My best regards!

Round 4

Reviewer 1 Report

Comments and Suggestions for Authors

Thank you for the answers. Please, next time use kinetics. It is not scientifically to use one (final) time point for measurement. The main changes occurred during elongation, not at statinary phase. This was the main mistake in biological design. Next6 time you need to do biologically relevant design of experiments.

Line 99: "Coumarin promotes hypocotyl " ???

Line 264: "Hypocotyl development and elongation depend dramatically on environmental conditions. " ???

My best regards!

Author Response

Comments 1:Thank you for the answers. Please, next time use kinetics. It is not scientifically to use one (final) time point for measurement. The main changes occurred during elongation, not at statinary phase. This was the main mistake in biological design. Next6 time you need to do biologically relevant design of experiments.
Response 1:Thanks a lot! In the upcoming research, we will definitely use kinetics.

Comments 2:Line 99: "Coumarin promotes hypocotyl " ???
Response 2:Corrected. Coumarin promotes hypocotyl elongation by promoting cell expansion

Comments 3:Line 264: "Hypocotyl development and elongation depend dramatically on environmental conditions. " 
Response 3:Changed to “The elongation of the hypocotyl is strongly influenced by the surrounding environmental conditions”